# Agroecological Management of the Grey Mould Fungus *Botrytis cinerea* by Plant Growth-Promoting Bacteria

**DOI:** 10.3390/plants12030637

**Published:** 2023-02-01

**Authors:** Ma. del Carmen Orozco-Mosqueda, Ajay Kumar, Ayomide Emmanuel Fadiji, Olubukola Oluranti Babalola, Gerardo Puopolo, Gustavo Santoyo

**Affiliations:** 1Departamento de Ingeniería Bioquímica, Tecnológico Nacional de México en Celaya, Celaya 38010, Gto, Mexico; 2Centre of Advanced study in Botany, Banaras Hindu University, Varanasi 221005, India; 3Food Security and Safety Focus Area, Faculty of Natural and Agricultural Sciences, North-West University, Private Bag X2046, Mmabatho 2735, South Africa; 4Center Agriculture Food Environment (C3A), University of Trento, Via Edmund Mach 1, 38098 San Michele all’Adige, Italy; 5Instituto de Investigaciones Químico-Biológicas, Universidad Michoacana de San Nicolás de Hidalgo, Morelia 58030, Mich, Mexico

**Keywords:** rhizobacteria, botrycides, sustainable agriculture, biocontrol mechanisms

## Abstract

*Botrytis cinerea* is the causal agent of grey mould and one of the most important plant pathogens in the world because of the damage it causes to fruits and vegetables. Although the application of botrycides is one of the most common plant protection strategies used in the world, the application of plant-beneficial bacteria might replace botrycides facilitating agroecological production practices. Based on this, we reviewed the different stages of *B. cinerea* infection in plants and the biocontrol mechanisms exerted by plant-beneficial bacteria, including the well-known plant growth-promoting bacteria (PGPB). Some PGPB mechanisms to control grey mould disease include antibiosis, space occupation, nutrient uptake, ethylene modulation, and the induction of plant defence mechanisms. In addition, recent studies on the action of anti-*Botrytis* compounds produced by PGPB and how they damage the conidial and mycelial structures of the pathogen are reviewed. Likewise, the advantages of individual inoculations of PGPB versus those that require the joint action of antagonist agents (microbial consortia) are discussed. Finally, it should be emphasised that PGPB are an excellent option to prevent grey mould in different crops and their use should be expanded for environmentally friendly agricultural practices.

## 1. Introduction

*Botrytis* spp. is a group of fungi (*Hyphomycetes*) characterised by several plant pathogenic species that cause serious damage to crops [1,2]. Almost 600 vascular plant genera, representing approximately 1400 plant species, can be infected by any of the 28 *Botrytis* species [3]. However, the *Botrytis cinerea* Persoon: Fries (teleomorph *Botryotinia fuckeliana* (from Bary) Whetzel), is considered the most destructive among all the species belonging to the genus *Botrytis*. This classification was derived from a survey of almost 500 members of the scientific community worldwide and pointed out that *B. cinerea* is behind *Magnaporthe oryzae*, a filamentous ascomycete fungus that is the causal agent of rice blast disease, the most destructive disease of rice worldwide [4]. *B*. *cinerea* is the causal agent of grey mould in more than 200 dicot plants, attacking tissues, such as stems, leaves, and fruits, as a necrotroph [5]. *B. cinerea* can affect economically important crops, such as vegetables (e.g., tomato, cucumber, and lettuce), ornamentals (e.g., rose and gerbera), bulbs (e.g., onion and ginseng), and fruits (e.g., grapevine, watermelon, and kiwifruit) [6,7,8]. In particular, *B. cinerea* can easily infect berries, such as strawberries, blueberries, raspberries, cranberries, and bilberry fruits, causing drastic losses after harvest [9,10].

In economic terms, losses in crops caused by *B. cinerea* can be difficult to estimate because of its wide range of host plants. However, they can be estimated to be worth millions of dollars or euros annually, depending on the agricultural sector. Some authors have estimated that the losses could be up to EUR 1 billion per annum if aspects such as cultural measures of pest control, botryticides, broad-spectrum fungicides, and biocontrol are included. Likewise, the infections caused by the fungal pathogen can vary, but it has been estimated that they cause between 10 and 70% of the losses pre- and post-harvest [3,4].

Even today, the use of chemicals is the main method used to control plant diseases both at pre- and post-harvest. Fungicides used strictly to control *B. cinerea* cover 10% of the global fungicide market [11]. To control plant diseases caused by *B. cinerea*, several families of synthetic botrycides are used, such as dichlofluanid and thiram, which are older and have a broad spectrum of action, as well as newer and more specific agents, such as fluazinam, boscalid, carbendazim, diethofencarb, dicloran, iprodione, procymidone, and vinclozolin, among others. These botrycides can be classified according to their mode of action into five categories as follows: (1) fungicides affecting fungal respiration; (2) anti-microtubule toxicants; (3) compounds affecting osmoregulation; (4) fungicides whose toxicity is reversed by amino acids; and (5) sterol biosynthesis inhibitors [9,12,13]. However, even when there is a wide variety of botrycides with different modes of action, the presence of resistant *B. cinerea* strains may occur, as this fungus may generate and accumulate mutations in its genome that allow its survival in the environment, resulting in relevant damages to crops around the world [14]. In addition to resistance to chemical botrycides, consumers prefer organic products that do not include the use of pesticides during their production. Producers have also observed this when trying to market their products abroad, where they must pass certain phytosanitary standards [15]. Another negative aspect of botrycides and pesticides, in general, is their extensive contamination of various environments, whether terrestrial, aerial, or aquatic, affecting biodiversity. Likewise, toxicity is not exempt in humans, causing various diseases [16].

Therefore, alternatives have been sought to eliminate or reduce the use of synthetic chemicals to control *B. cinerea*, including the use of microbial agents such as *Trichoderma harzianum* [17], *T. viride*, *T. virens* [18], *Ulocladium* spp. [19], *Clonostachys rosea* [20], *Gliocladium catenulatum*, *Saccharomyces cerevisiae*, *Wickerhamomyces anomalus*, *Metschnikowia pulcherrima*, and *Aureobasidium pullulans* [21,22], as well as diverse plant growth-promoting bacterial species (PGPB) [11,23,24,25,26,27]. This last group of beneficial bacteria, in addition to stimulating plant growth directly, can also antagonise fungal pathogens, such as *B. cinerea,* through various mechanisms [28]. Some of the mechanisms of bacterial plant growth stimulation include providing plants with nutrients that are difficult to obtain directly from soil (or the atmosphere), such as nitrogen fixation, sequestering iron, or zinc and phosphate solubilization. PGPB can also modulate plant hormone levels, including indole-3-acetic acid, cytokinins, gibberellins, or ethylene. The plant hormone ethylene has a wide range of biological activities and is synthesized as a response to various stresses [29,30]. This review analyses the fundamental role of this group of microbial agents in the biological control of *B. cinerea*, the cause of grey mould disease.

## 2. *Botrytis cinerea* Life Cycle and Infection Stages

*B. cinerea* is considered a hemibiotrophic fungus, since, during some short stages of its life cycle, it can act as a biotrophic fungus and colonise living plant tissues and obtain nutrients from living plant cells [31]. In contrast, some authors consider *B. cinerea* among the pathogenic fungi that are necrotrophic since it also infects and kills plant tissues and subsequently extracts nutrients from dead plant cells. Similarly, there is evidence that *B. cinerea* and other *Botrytis* species are too versatile to be captured in a single category [32]. This is, in a way, understandable, but it is not clear how it infects more than a thousand plant species.

Regarding the *B. cinerea* infection cycle, van Kan and Shaw [33] identified and listed a series of stages, as follows: (1) the attachment of the conidia; (2) germination; (3) differentiation of infection structures on the host plant surface; (4) penetration of the host plant tissues; (5) killing the host plant; (6) formation of primary lesions and avoiding host plant defences; (7) disease expansion and tissue maceration. At the first stage, the attachment of the conidia involves mechanical adhesion forces and hydrophobic interactions with the host plant cells, as well as the formation of a matrix of fungal enzymes that could help avoid early defences of the host plant and prevent dehydration of the *B. cinerea* hyphae. At the second stage, the conidia require moisture to germinate, which is essential for subsequent infection, after which it reaches the third stage, where the differentiation and formation of appressoria occur during penetration (fourth stage). Once it penetrates the host plant tissue, *B. cinerea* kills the host plant cells before they are invaded by hyphae. An indicator of cell death is the activation of nuclear condensation and damage to plant membranes, which is a relevant step for infection [34]. At the penultimate stage, *Botrytis* is dedicated to necrotising the tissue, stimulating a series of responses in the host plant, such as the production of reactive oxygen species (ROS). However, if the fungus manages to avoid them, it can reach the last phase of the infection cycle and spread the disease in various tissues or fruits [33]. The life cycle of *B. cinerea* is also a part of grey mould disease and depends on different environmental conditions, such as humidity and temperature, that vary during the seasons of the year [35]. Figure 1 describes the essential steps of the grey mould disease cycle caused by *B. cinerea*.

## 3. Antifungal Mechanisms of Plant Growth-Promoting Bacteria (PGPB)

PGPB include all the bacterial communities that are associated with the plant, inhabiting different compartments, and exerting a stimulating effect on their growth through different direct and indirect mechanisms [36,37]. PGPB can reside belowground and exert their beneficial effects from the rhizosphere (rhizobacteria), defined as the soil that surrounds the root and where root exudates exert an effect on the inhabiting microbiota. The rhizosphere bacterial communities are among the most studied bacterial communities and there is a large amount of literature on their extensive benefits in agriculture [30,38,39,40,41,42]. Another area where bacterial communities reside and associate is the phyllosphere (phyllosphere or epiphytic bacteria), the above-ground surface of plants. The phyllosphere is also a complex ecosystem, in which bacteria (and many other microbes) interact extensively and play important roles, including protection against plant pathogens [36,43]. Another zone of close interaction is the endosphere, which includes below- and aboveground plant parts, which can be divided into internal tissues, including roots, stems, leaves, fruits, seeds, and flowers [44,45,46]. Some authors suggest that endophytic bacteria exert a closer interaction than the microbiota that resides outside the plant [37]. For example, the production and excretion of hormones by an endophytic bacterium, regardless of the concentration, is directly perceived by the plant. On the contrary, the same amount of hormone emitted by an outside bacterium could volatilise and lose its effect on the plant, as it does not reach a detectable concentration in plant tissues [47,48]. However, this hypothesis requires strong and convincing evidence.

It can be hypothesised that the bacteria that reside in the rhizosphere defend the plant more forcefully through the synthesis of antibiotics before the pathogen approaches and colonises the plant surface. PGPB are excellent microbial biocontrol agents that play an important role in the antagonism and control of fungal plant pathogens [30,49,50], such as *B. cinerea* [24,51,52].

In general, PGPB can antagonise, inhibit, or kill fungal plant pathogens through different mechanisms, such as antibiosis, the production of lytic or cell wall-degrading enzymes, siderophores, competition and occupation of spaces, lowering the amount of ethylene, and triggering the induced systemic resistance (ISR). Several excellent review papers have described these mechanisms in detail [49,53,54,55,56]. Some of these will be mentioned only briefly in this paper.

### 3.1. Antibiosis

Antibiosis includes the production of antibiotics, including bacterial secondary metabolites, such as 2,4-diacetylphloroglucinol, bacillomycin, fengycin, herbicolin, iturin, oomycin, phenazine-1-carboxylic acid, pyoluteorin, pyrrolnitrin, surfactin, tension, and viscosinamide [57,58,59]. Other types of antibiotic compounds include volatile organic compounds (VOCs), whose characteristic, as their name implies, is to volatilise in the environment [60]. Some examples of antifungal VOCs include 2-ethyl 1-hexanol, benzothiazole, cyclohexanol, dimethylhexadecylamine, dimethyl disulfide, dimethyl trisulfide, hydrogen cyanide, n-decanal and, nonanal [61,62,63], to mention but a few.

### 3.2. Siderophores and Space Occupation

The production of siderophores, iron-chelating compounds, is one of the first mechanisms involved in the inhibition of plant pathogen growth [64]. Bacteria able to produce them (e.g., *Pseudomonas* spp.) can deprive iron plant pathogens in ecosystems, such as the rhizosphere, and thus occupy spaces. At the same time, bacteria displace potential plant pathogens from these sites, including nutrient-rich sites close to the plant roots [65,66].

In comparison to bulk soil, the rhizosphere is a nutrient-rich environmental niche for microbes, including PGPB. Root exudates include nutrients such as organic acids, sugars, and amino acids [39]. Therefore, acquiring these nutrients becomes an act of competition, in addition to occupying the best spaces or those closest to the plant roots. The mechanism of competition and occupation of spaces has also been described in various PGPB, including those of rapid growth, such as *Bacillus* (*Ba.*) *licheniformis* species, and *Pseudomonas fluorescens*, which can metabolise a vast array of nutrients, among other complex compounds [67,68]. Additionally, *Bacillus* spp. and *Pseudomonas* spp. are usually excellent biofilm producers, and thus, they firmly bind the plant roots, where they can exert their beneficial activities, such as the synthesis of siderophores [69,70,71,72].

### 3.3. Ethylene and 1-aminocyclopropane-1-carboxylate (ACC) Deaminase Activity

When plants are subjected to a variety of abiotic and biotic stresses, such as a plant pathogen attack, the ethylene levels increase, causing various deleterious effects in tissues and slowing their growth [73,74,75]. Most PGPB are producers of the enzyme ACC deaminase [76]. ACC deaminase reduces ethylene levels by degrading the ACC precursor. Numerous studies have been carried out on the action of ACC deaminase responsible for the biocontrol activity of plant pathogens in various PGPB [45,77,78,79,80].

### 3.4. Induced Systemic Resistance

The so-called systemic acquired resistance (SAR) is triggered by the attack of plant pathogens, whereas the induced systemic resistance (ISR) is stimulated by non-pathogenic bacteria, including various species of PGPB [81,82]. This ISR mechanism is important because it stimulates the expressions of genes involved in strengthening the plant immune system, in addition to inducing tissue lignification and increasing peroxidase and superoxide dismutase activity within the treated plant [83,84].

In contrast, SAR is mainly characterised by the coordinated expression of multiple genes related to resistance to pathogenesis, better known as pathogenesis-related (PR) genes. PR genes encode proteins with broad antimicrobial activity [85,86]. In this way, the activation of genes such as PR1, PR2, and PR5 depends on salicylic acid (SA) signalling, while PDF1.2, PR3, and PR4 are activated through an SA-independent and jasmonic acid (JA)/ethylene (ET)-dependent pathway [84]. In a study where *Arabidopsis* plants were pre-treated with the plant-beneficial *B. cereus* strain AR156, it stimulated the expressions of PR1, PR2, PR5, and PDF1.2, suggesting that both the SA and JA/ET signalling pathways were activated [87]. This study and other pioneering works have shown that NPR1 is relevant for coordinating both signalling pathways [83,88].

In a recent work, grapevine plants inoculated with *Burkholderia* strains BE17 and BE24 accumulated more reactive oxygen species and an increased callose deposition compared to uninoculated controls as a response to *Botrytis cinerea* infection. Additionally, bacterized plant leaves also overexpressed pathogenesis-related (PR) proteins (*PR5* and *PR10*) and other two markers involved in the SA-signalling pathway [89].

Vegetable crops such as grapevine, tobacco, tomato, strawberry, bean, pepper, and cucumber, among others, have been stimulated in their defensive responses, including functions such as ascorbate peroxidase, catalase, glutation peroxidase, lipid hydroperoxidase, lipoxygenase, plant defensin, peroxidase, pathogenesis-related proteins, and superoxide dismutase, by multiple bacteria of genera, such as *Bacillus*, *Burkholderia*, *Pseudomonas*, or *Streptomyces* [27].

## 4. Antagonistic Effects of PGPB on *Botrytis cinerea*

The metabolites produced by PGPB, such as *Pseudomonas* spp., can vary and affect the initial or intermediate stages of the *B. cinerea* life cycle and its infection process. For example, *P. antimicrobica* inhibits the germination of conidia in *B. cinerea* by producing compounds that affect the formation of abnormal germ structures and reducing the extension and morphology of the germ tube [90]. Other studies have revealed that the plasma membrane of *B. cinerea* conidia can be severely damaged by antifungal compounds in the supernatant of *P. fluorescens* strain QBA5 [91]. Likewise, *Pseudomonas* spp. strains produce lipopolysaccharides with an important function to control the growth of the *B. cinerea* mycelium by damaging the hyphae and distorting their morphology [92].

The *Bacillus* genus groups multiple species that have a wide diversity of mechanisms to inhibit the growth of pathogens such as *B. cinerea*. For example, the *Bacillus amyloliquefaciens* group, which includes species such as *B. amyloliquefaciens*, *B. velezensis*, *B. nakamurai*, and *B. siamensis*, has been very effective in postharvest action to reduce the harmful effects of *Botrytis*, among other fungal pathogens [93].

Other Bacilli species, such as the *Ba. cereus* strain B-02, isolated from the tomato rhizosphere, can also suppress the grey mould in tomato caused by *B. cinerea*. The antagonism mechanisms cause changes in the morphology, ultrastructure, and physiology of hyphae and inhibit *B. cinerea* spore germination [94]. Because *Bacillus* spp. produce a relevant diversity of lytic enzymes that degrade the fungal cell wall, such as proteases, lipases, chitinases, and glucanase, and different antibiotics, it is possible that such an arsenal directly affects the germination of the spores and the normal growth of the mycelium of *B. cinerea* [25,95,96]. This has been corroborated by Ait Barka et al. [97], who observed direct contact between *Burkholderia phytofirmans* PsJN and the mycelium of *B. cinerea*, noting a growth disruption in the fungal mycelium, coagulation, and protoplasm leakage. Interestingly, the authors also evaluated the effect of bacteria on endopolygalacturonase activity on a solid medium; however, no effect was observed. Therefore, the authors concluded that strain PsJN inhibits the growth of *B. cinerea* by disrupting the cellular membrane and inducing cell death. Figure 2 shows the potential effects of PGPB at different growth stages of *B. cinerea*.

## 5. Biological Control of *Botrytis cinerea* by PGPB

In different parts of the world, the main method of controlling grey mould infections is the use of chemical botrycides, which is a multi-million dollar market [4]. However, the secondary effects that they have on the environment and human health have been widely reported, particularly in producers who do not use adequate means of protection to avoid inhaling these [16]. As mentioned in the introduction, Leroux [9] classified various botrycides according to their modes of action, such as those that affect fungal respiration, anti-microtubule toxicants, compounds affecting osmoregulation, fungicides whose toxicity is reversed by amino acids, and sterol biosynthesis inhibitors. Likewise, the resistance of some *Botrytis* spp. strains continues to cause damage and economic losses [5].

Thus, the development of new strategies to control infections caused by *B. cinerea* is urgently warranted in agriculture, including those based on preventive action [98]. In this sense, Escribano-Viana et al. [99] prophylactically applied a biofungicide containing the agent *Ba. subtilis* QST713 to vineyards (*Vitis vinifera* L. cv. Tempranillo) in Spain, and its effect was compared with that of a chemical fungicide composed of fenhexamid regarding oenological parameters. The results showed that the application of *Ba. subtilis* QST713 did not affect the grapes or the quality of the wine, and the protective effects were similar to those of the chemical fungicide. Both had a positive influence on grape production in vineyards. Interestingly, the application of the bioinoculant had a positive effect on the populations of *Saccharomyces cerevisiae* with different genotypes by improving the implementation of malolactic fermentation. These results suggest that the use of biofungicides is a viable and eco-friendly strategy to control grey mould in vineyards, where the grape is particularly susceptible to *Botrytis* infections, and that it does not interfere with oenological parameters.

In addition to bacteria associated with grapevines, various species of PGPB have been isolated and characterised, such as *P. fluorescens* PTA-268 and PTA-CT2, *Ba. subtilis* PTA-271, *Pantoea agglomerans* PTA-AF1 and PTA-AF2, and *Acinetobacter lwoffii* PTA- 113 and PTA-152, which were characterised as possible new biological control agents against *B. cinerea*. Some of these species exhibit a dual antifungal effect, which includes direct antagonism, and the induction of defence responses in plants. Some of the mechanisms stimulated in the grapevine leaves include the activities of lipoxygenase, phenylalanine ammonia-lyase, and chitinase, which protect leaves from infection by *B. cinerea* [100].

Another recent study that compared the in vitro antagonistic effects of VOCs, such as dimethylhedecylamine (DMHDA) produced by the PGPR *Arthrobacter agilis* UMCV2, with the broad-spectrum fungicide captan, was carried out by Velázquez-Becerra and colleagues [101]. An evaluation of the effect of similar compounds, that is, aminolipids containing 4, 8, 10, 12, and 14 carbons in the alkyl chain, showed that DMHDA had a greater antifungal effect, which was comparable to that of captan. Interestingly, DMHDA inhibited the growth of *B. cinerea* mycelium, the plant pathogenic oomycete *Phytophthora cinnamomi*, and showed a minor inhibitory effect against beneficial fungi, such as *Trichoderma* spp. These results suggest that this VOC has a specific antifungal action against potential plant pathogens, but not against other plant-beneficial microorganisms in the rhizosphere.

Fruit berries are particularly susceptible to *B. cinerea* attack, and in the case of grapevine, it is the greatest threat as it causes economic losses [3]. Recently, the effects of temperature and relative humidity on *B. cinerea* infection in grape berries were evaluated, as were the protective effects of six microbial biocontrol agents, including *Aureobasidium pullulans*, *Ba. amyloliquefaciens*, *Ba. amyloliquefaciens* subsp. *plantarum*, *Ba. subtilis*, *Pythium oligandrum*, and *T. atroviride*. The biocontrol agents mentioned above are part of commercial inoculants, such as *Ba. amyloliquefaciens* strain FZB24, which is present in the product named Taegro (TAE), marketed by Syngenta, or *Ba. subtilis* QST 713, with the product name Serenade max (SER), marketed by Bayer SpA; the results of this study showed that environmental conditions (humidity and temperature) influenced the success rate of the biocontrol agents against fungal damage [102]. The aforementioned bacterial agents (e.g., *Ba. subtilis* QST 713) are widely known for their biocontrol capabilities against fungal pathogens and their excellent growth-stimulating ability in plants, such as canola (*Brassica campestris*) [103], cucumber (*Cucumis sativus* L.) [104], and tomato (*Solanum lycopersicum* L.) [105]. In the case of PGPB *Ba. amyloliquefaciens* FZB24, its protective effects have been observed in plants, such as tomato (*Solanum lycopersicum* L.) [106], and it has stimulating effects on cotton production [107] and the growth of *Lemna minor* [108]. An excellent review of beneficial indirect mechanisms—including pathogen biocontrol—of the FZB24 strain is available [55].

Different strains of *Bacillus* species are, in general, widely isolated and characterised by their broad antifungal activities (among other plant growth-stimulating effects), as shown by Jian and colleagues [109], who demonstrated that two *Ba. velezensis* strains, 5YN8 and DSN012, could significantly control pepper grey mould disease and promote pepper (*Capsicum frutescens*) plant growth. The action of secondary (diffusible) metabolites or the emission of VOCs, which suppress the growth and spore formation of *B. cinerea*, could be among the biocontrol mechanisms of *Ba. velezensis* strains. Finally, *Ba. velezensis* strains were also found to induce the immune mechanisms of pepper plants, revealed via the expression profiling of genes, such as *NPR1, PR1, PIN2, TIN1*, and peroxidase-coding genes in leaves.

Another recent study [110] demonstrated the protective effect of the *Ba. velezensis* strain XT1 against *B. cinerea* in strawberry and tomato plants. *Ba. velezensis* XT1 was efficient in activating the defence response through plant phytohormonal regulation via two application methods—foliar and root. In particular, foliar applications of *Ba. velezensis* XT1 only led to elevated levels of antifungal action, whereas root applications of this bacterial strain increased both plant biomass and protection against *B. cinerea*. Likewise, *Ba. velezensis* XT1 also produces cyclic lipopeptides with antifungal activity against *B. cinerea* [110].

Previous works have shown that the application method can influence the biocontrol efficacy of the bacterial agent, and as described throughout the text, the mechanisms of the action of PGPB may differ. For example, Salvatierra-Martinez et al. [51] recently evaluated the ability of *Ba. amyloliquefaciens* BBC023 and BBC047 to colonise plant tissues and control *B. cinerea* in the Philippines. The authors concluded that both strains, BBC047 in particular, showed better abilities for phyllosphere colonisation, which was also related to the better control of the fungus. Likewise, both *Ba. amyloliquefaciens* strains were efficient in stimulating the growth of tomato plants.

The antifungal action is not only exclusive to *Bacillus* spp., although there are abundant reports on *Bacillus* species. Other PGPB species, such as *Pseudomonas* spp., have also exhibited multiple antagonistic effects that reduce the growth, conidia germination, and infection of *B. cinerea* [25,90,103]. This is the case for the *P. fluorescens* UM16, UM240, UM256, and UM270 strains, which produce diffusible compounds (phenazines, cyanogens, and ACC deaminase), biofilm, siderophores, proteases, and indole-3-acetic acid, along with volatiles (dimethyl disulfide, dimethylhexadecylamine, and hydrogen cyanide), whose action reduces disease symptoms in *Medicago truncatula* plants [24].

Upon extensive screening of *Pseudomonas* spp. strains, Mikani et al. [111] selected ten strains that showed excellent in vitro and in vivo activity to control grey mould on apple fruits, with inhibition ranges that varied depending on the assay performed. For example, it ranged from 49 to 68% in dual culture tests, from 75 to 99% in cell-free culture filtrate tests, and from 52 to 97%in VOC tests. In fruit trials, the strains also antagonised the pathogen, highlighting the *P. fluorescens* strain Pf1 as the most efficient.

## 6. PGPB Consortia to Control *Botrytis cinerea*

Plants are in constant contact with microorganisms that inhabit plant tissues, including the rhizosphere, phyllosophere, and endosphere [23,112,113]. Thus, the interactions between plants and microorganisms, including PGPB, are usually intensive. One of the strategies to improve the effectiveness of the biocontrol of plant pathogens, such as *B. cinerea*, is to apply a consortium of two or more strains belonging to PGPB species [114], or even mixtures of bacteria with other microbial groups, such as *Trichoderma* or arbuscular mycorrhizal fungi (AMF) [115,116,117,118] (Table 1). A previous study showed the inoculation of antagonists, such as *T. atroviride*, *A. pullulans*, and *Ba. subtilis*, over four years in three locations in Italy, which were applied at bunch-closure, veraison, and pre-harvest, respectively, to control *B. cinerea* on grapevine bunches. The results were significant when a mixture of the microbial antagonists was applied; however, the results were equally successful when each microorganism was applied separately. The authors suggest that the high level of efficacy of the tested biocontrol agents against grey mould disease can be explained by the integration of good agronomic practices, a relatively medium-low level of the disease, and the optimal timing of the treatment [115].

Using beneficial microbes, such as PGPB, in bioformulations for efficient crop productivity and protection presents an efficient and environmentally friendly alternative to the agrochemicals used today [119]. Therefore, detecting the main action mechanisms in PGPB is a relevant strategy for developing better bioinoculant mixtures, which could compensate for the lack of biocontrol or direct plant growth-promoting mechanism in a strain or species of the applied consortia. For example, in a recent study, different strains of *Streptomyces* spp. (namely strain ATIRS43, ATIRS65, and ARRS10) showed excellent inhibitory activity and protection against *B. cinerea* in pea plants. However, pea plants have a specific symbiotic relationship with nodular and nitrogen-fixing species of rhizobia, and finding antagonistic or antifungal activities in these strains is not a common characteristic. Therefore, the co-inoculation of the three *Streptomyces* spp. strains with *Mesorhizobium ciceri* UPM-Ca7T increased the nodulation and nitrogenase activity in five chickpea genotypes, namely ICCV2, ICCV10, ICC4958, Annigeri, and JG11. Likewise, the *Streptomyces* spp. strains significantly reduced diseases caused by *B. cinerea* by 28–47% compared to uninoculated chickpea plants [120].

Another example of the synergistic or additive action of the application of PGPB consortia is the work of Rojas-Solís et al. [121]. In this work, the authors co-inoculated two strains of PGPB (*P. stutzeri* E25 and *Stenotrophomonas maltophilia* CR71), which promoted the chlorophyll content, shoot and root length, and total fresh weight of tomato plants (*Solanum lycopersicum* cv. Saladette) when introduced in a co-inoculation. Additionally, both strains grew without any counteraction in a co-inoculation, where both strains were present in the in vitro growth medium. Finally, both strains also exhibited excellent biocontrol mechanisms towards *B. cinerea*, including the production of VOCs, such as dimethyl disulfide [121].

Another strain of *Ba. thuringiensis* (UM96), with good chitinase activity [25], which is important for inhibiting the mycelial growth of *B. cinerea*, presented good rhizosphere colonisation capacity and interaction with *P. fluorescens* strains. For example, the UM96 + UM256 mixture showed significant improvements in promoting the growth of husk tomato plants (*Physalis ixocarpa* Brot. Ex Horm.) [122]. The inoculation results from the same bacterial consortium showed an increase in the protection of husk tomato plants against *B. cinerea* infection (Santoyo et al., unpublished results).

However, not all possible mixtures are usually successful in stimulating plant growth or protecting plants from infection, as observed upon the individual or combined application of two *Pseudomonas* (LBUM223 and WCS417r) and two *Bacillus* strains (LBUM279 and LBUM979) for root treatments in cannabis seedlings, which were subsequently infected with *B. cinerea*. The results did not show any significant control of grey mould, and all infected leaf tissues were necrotic after a week, regardless of the treatment. Similarly, when evaluating whether rhizobacteria could induce possible cannabis defence-related genes, no positive results were obtained [123]. In addition, Vijayabharathi et al. [124] observed that the inoculation of a consortium of endophytic *Streptomyces* spp. strains was less effective than individual inoculation in controlling grey mould disease caused by *B. cinerea* in chickpeas with different genotypes. The results showed that not all consortia were more effective with better activities than single inoculations, either through synergistic or additive actions. Therefore, there may be competition for a space in the rhizosphere between the inoculated strains, as well as some antagonism. For instance, some VOCs produced by beneficial bacteria can modulate the growth and motility of other plant or bacteria species [62]. For this reason, it is advisable to perform in vitro confrontational tests to see if there is antagonism between the mixtures of microorganisms that are to be applied as a consortium [122], avoiding unwanted results in the field. A review on the stimulation of plant growth by microbial consortia was recently published, discussing different inoculation strategies and synergistic action mechanisms between microorganisms (PGPB, *Trichoderma* spp., and AMF) associated with plants [114].

**Table 1 plants-12-00637-t001:** Examples of works reviewed highlighting the action mechanisms and antagonistic effects exerted against the grey mould phytopathogen *B. cinerea*. Some works also show the dual activity of plant protection and stimulation of plant growth by plant growth-promoting bacteria. ND means not determined.

Biocontrol Agent	Protected Host	Mechanisms of Action Exerted	Antagonistic Effect on *B. cinerea*/Benefit on Plant	Reference
*Pseudomonas antimicrobica*	None	Antifungal metabolites	Affectations on germ tube production and extension	[90]
*P. aeruginosa* strain LV	None	Phenazine-1-carboxylic acid (PCA) produced	Damage on the hyphae; mycelial growth inhibition	[92]
*P. fluorescens* strain QBA5	Tomato fruit and plant leaves (cv. Laifen No.1)	Supernatant bioactive compounds	Damage in the conidia germination and plasma membrane; plant and fruit ripening protection	[91]
*Bacillus cereus* strain B-02	None	Supernatant bioactive compounds	Changes on cell morphology (distortion, shrinking, and swelling)	[94]
*Pseudomonas* sp. strain PsJN	Plantlets of *V. vinifera* L. ‘Chardonnay’	Diffusible antagonistic compounds	Growth disruption of fungal mycelium, coagulation, and leakage of protoplasm; plant protection	[97]
*Bacillus subtilis* strain QST713	*Vitis vinifera* L. cv. Tempranillo	ND	ND; positive influence on grape production and oenological parameters	[99]
*P. fluorescens* PTA-268 and PTA-CT2, *Bacillus subtilis* PTA-271, *Pantoea agglomerans* PTA-AF1 and PTA-AF2, and *Acinetobacter lwoffii* PTA- 113 and PTA-152	*Vitis vinifera*	Induced systemic resistance (ISR)	ND; stimulation in the leaves of vine plants included the activities of lipoxygenase, phenylalanine ammonia-lyase, and chitinase	[100]
*Arthrobacter agilis* UMCV2	None	Dimethylhexadecylamine (DMHDA)	Mycelial growth inhibition	[101]
*Aureobasidium pullulans*, *Bacillus amyloliquefaciens*, *Bacillus amyloliquefaciens plantarum*, *Bacillus subtilis*, *Pythium oligandrum*, and *Trichoderma atroviride*	Grape berries	ND	ND; reduction on *Botrytis* bunch rot (BBR) disease	[102]
*B. velezensis* strains, 5YN8 and DSN012	Pepper (*Capsicum frutescens*)	Secondary (diffusible) metabolites and volatile organic compounds (VOCs)	Suppression of the growth and spore formation; plant growth promotion	[109]
*B. velezensis* strain XT1	Tomato and strawberry plants	Direct foliar and radicular application	ND; activation of the defence system through phytohormonal regulation	[110]
*B. amyloliquefaciens* strains BBC023 and BBC047	Tomato plants	Phyllosphere colonization good capacity	ND; stimulation of tomato plant growth	[51]
*P. fluorescens* strains UM16, UM240, UM256, and UM270	*Medicago truncatula* plants	Potential diffusible compounds (phenazines, cyanogens, and ACC 1-aminocyclopropane-1-carboxylate deaminase, production of biofilm, siderophores, proteases, indole-3-acetic acid), and volatiles such as dimethyl disulfide, dimethylhexadecylamine, and hydrogen cyanide	Promotion of *Medicago truncatula* plant biomass and chlorophyll content	[24]
*Bacillus subtilis*, *Trichoderma atroviride*, and *Aureobasidium pullulans*	Commercial vineyards in three locations in Italy	Direct application of mixture of biocontrol microorganisms	Biocontrol of the grey mould disease	[115]
*Streptomyces* spp. strains AUR2, AUR4, and ARR4, *Mesorhizobium ciceri*	Chickpea plants	Single and mixed inoculation *in planta*	Biocontrol of the grey mould disease; plant protection and induction antioxidant enzymes; enhanced nodulation and nitrogenase activity	[124]
*Streptomyces* spp. strains ATIRS43, ATIRS65, and ARRS10	Chickpea plants; marigold (*Tagetes erecta* L.) flower	Potential action of HCN, ammonia (except ATIRS65), ß-1,3-glucanase, chitinase, cellulase (except ATIRS 65), protease, lipase, and siderophores (except ATIRS65)	Reduction on the grey mould disease incidence; stimulation of the plant growth and flower number	[120]

## 7. Biocontrol Strategies to Prevent *Botrytis cinerea* Infection

The ideal way to control *B. cinerea* infections is to avoid possible plant infections before the harvest. However, plant maintenance tasks, such as aerial tissue irrigation, increase humidity results in more favourable conditions for the establishment of *B. cinerea* through the conidia. Therefore, to control the infection, different strategies have been proposed, including avoiding the attachment of the conidium to the host plant by modifying the surface properties [3]. It has been observed that bacteria, such as *Bacillus* spp. and *Pseudomonas* spp., can change the leaf wettability [3,125,126]. Likewise, the foliar application of bioinoculants can protect and antagonise the germination of conidia before adhering to the host plant tissue. Therefore, it is important to apply PGPB, which inhibit the germination of *B. cinerea* conidia [90]. PGPB and their antifungal mechanisms may play an essential role since once the host plant tissue is necrotised, the damage is irreversible. In the case of post-harvest infections in fruits, this late stage of *B. cinerea* infection is unfortunate, since the aesthetics and quality of the product are affected [127]. Thus, the prophylactic protection of crops is important for controlling grey mould disease. Identifying bacterial compounds that inhibit the attachment or specific antagonistic action against *B. conidia* requires further investigation.

If *B. cinerea* can adhere to and penetrate the host plant surface, it is also possible to contain a larger infection that could spread to other hosts. When this stage of infection occurs, some symptoms may not be visible [128]. The spray application of antagonistic agents could also help avoid further damage and possible infections in flowers and fruits post-harvest. PGPB also inhibit the growth of *B. cinerea* mycelia, and it is one of the main mechanisms that has been studied since it damages and deforms hyphal cells [54]. Multiple post-harvest studies have documented the protective effects of PGPB (as well as their compounds and lytic enzymes) on fruits, such as berries, which are particularly susceptible to attacks by *B. cinerea*. Likewise, the combination of natural compounds, such as chitosan (poly-D-glucosamine), or oligosaccharides of vegetable origin could be applied together with PGPB. It has been documented that some of these compounds can additionally inhibit the growth of *B. cinerea* and elicit plant defence responses [129,130,131]. Figure 3 illustrates two possible scenarios. One where a bioinoculant based on PGPB is applied preventively, which could lead to a beneficial interaction with the plant by inhibiting the growth of plant pathogens, stimulating plant growth, and resulting in good yield and better quality of fruits, and therefore, financial gains. The second scenario represents an uninoculated crop, prone to being infected by pathogens, with a high probability of having low production, low fruit quality, and economic losses.

The mixture of PGPB or their antifungal metabolites with other beneficial fungi, such as *Trichoderma* spp., can also be an excellent option to prevent host attachment, inhibit conidia germination, and avoid hyphae dispersion or their growth in flowers and fruits [132]. Some commercial bioinoculant products are available to control the infection before the establishment of the pathogen through foliar application in the field, such as Serenade (Agra Quest, Davis, CA, USA), which contains the *Ba. subtilis* strain QST 713. Other products, such as Bio-saves (Eco Science Corp., Cary, NC, USA), contain *P. syringae* and are used mainly for the post-harvest control of *B. cinerea* on different fruits [3]. Another biofungicide is Fungifree (Agro & Biotecnia, S. de R.L. de C.V.), based on *B. subtilis*, which is marketed mainly in Mexico [133]. However, according to the product’s technical data sheet, it also protects berry plants, such as strawberries, blueberries, and raspberries, and is sprayed in a liquid solution in a foliar form during pre-harvest. Table 2 shows available commercial biofungicides for the biocontrol of grey mould disease, based on antagonistic PGPB belonging to *Bacillus, Pantoea, Pseudomonas*, and *Streptomyces* genera.

## 8. Concluding Remarks

The genus *Botrytis* belongs to the family *Sclerotiniaceae*, which comprises fungal species with a worldwide presence, causing great damage to agricultural fields. In particular, the genus *Botrytis* contains approximately 30 recognised species with diverse trophic lifestyles and is considered among the top ten most important fungal pathogens [4]. According to van Kan et al. [31], the *B. cinerea* species is the best-studied member of this genus, which appears to display a facultative secretive endophytic behaviour (‘hide and seek’). Usually, when a microorganism, whether pathogenic or non-pathogenic, contains this genomic and phenotypic plasticity that allows it to explore different niches and adapt to different environmental conditions [134,135], its presence may be detected in different geographical and climatic regions of the world, including agricultural fields [136]. Here, we report that *B. cinerea* has excellent humidity and optimum temperature for its growth and infection, particularly under greenhouse conditions [16,137]. However, even when pre-treatment with botrycides is carried out, resistance in certain strains can continue to cause certain challenges for producers [10].

PGPB, therefore, arises as an emergency solution in agriculture to control *B. cinerea* infections [36,138,139]. It is essential to identify compounds that inhibit the early stages of grey mould infection, particularly to avoid host attachment. An interesting topic that would be good to delve into is the ability of PGPB to help plants recruit other beneficial microbiota, including those with the ability to restrict the growth of plant pathogens or pests [140]. Likewise, new strategies based on the combination of antifungal compounds from PGPB and the bioformulation of antifungal agents that contain bacterial agents could help reduce or avoid infections in plants caused by *B. cinerea*. In this same sense, the use of fungal microorganisms in combination with antagonistic bacteria can result in synergistic actions against *Botrytis cinerea*. However, there are still some gaps in the knowledge of bacteria-fungus interactions that can better direct their dual action in the field and that do not harm other endemic beneficial organisms. Other precautions that must be taken before releasing a biofungicide were recently proposed by other authors [22]. Finally, the search and characterisation of new species of PGPB and their anti-*Botrytis* compounds will continue to be a resource that should be explored with greater effort to have better and more sustainable agricultural practices for the sake of our planet.

## Figures and Tables

**Figure 1 plants-12-00637-f001:**
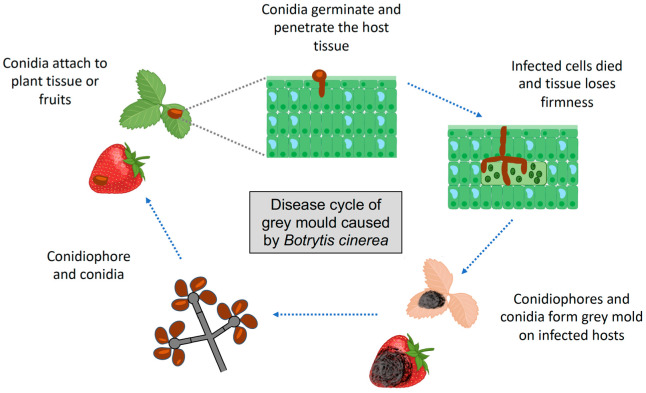
A summarised description of the disease cycle of grey mould caused by *Botrytis cinerea* in plant tissues and fruits.

**Figure 2 plants-12-00637-f002:**
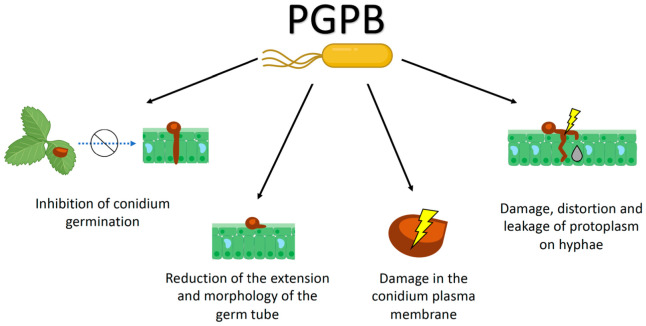
Effect of plant growth-promoting bacteria on different growth stages of *Botrytis cinerea*.

**Figure 3 plants-12-00637-f003:**
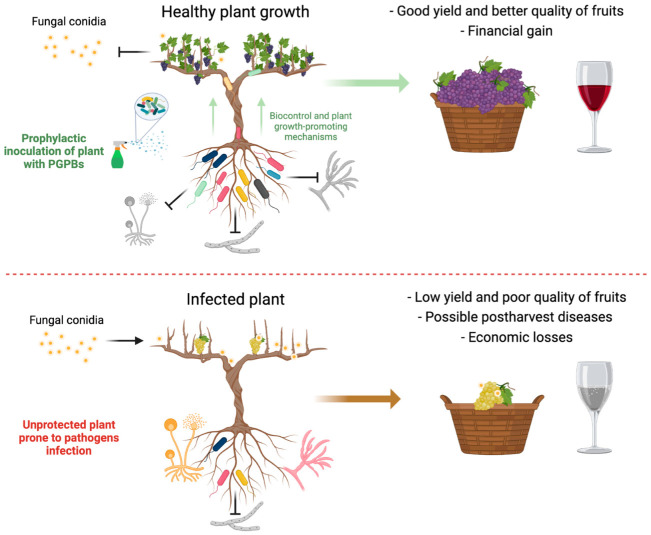
An illustration of two possible scenarios, depending on the preventive application or not of plant growth-promoting bacteria in crops (see text for details).

**Table 2 plants-12-00637-t002:** Available biofungicides based on antagonistic bacteria for the control of grey mould disease caused by *Botrytis cinerea* (and other potential fungal plant pathogens).

Bacterial Species/Strain	Trade Name ^®^	Company (and/or Country)
*Pantoea agglomerans*	Pantovital	IRTA (Spain)
*Bacillus subtilis*	Serenade Max	Bayer, formerly BASF (Germany)
*Pseudomonas syringae* strain ESC-10	Bio-save	Jet Harvest Solutions (USA)
*Bacillus amyloliquefaciens*	Amylo-X	Biogard CBC (Italy)
*Bacillus amyloliquefaciens*	Double Nickel 55WDG/LC	Certis (USA)
*Bacillus subtilis GB03*	Companion	Growth Products (USA)
*Bacillus subtilis IK-1080*	Botokira Wettable Powder	Idemitsu Kosan Inc., Japan
*Bacillus megaterium*	Bio Arc	Sphere Bio-Arc PVT Ltd. (India)
*Streptomyces griseoviridis* strain K61	Mycostop	Verdera Oy (Finland)
*Streptomyces lydicus* WYEC 108	Actinovate	Novozymes (Denmark)
*Bacillus subtilis*	Fungifree	Agro & Biotecnia, S. de R.L. de C.V. (México)
*Bacillus subtilis* strain QST 713	Serenade	AgraQuest (USA)
*Bacillus pumilus* strain QST2808	Sonata	AgraQuest, Davis-CA (USA)

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
