# Peer review of "Agroecological Management of the Grey Mould Fungus Botrytis cinerea by Plant Growth-Promoting Bacteria"

_plants, 2023, doi:10.3390/plants12030637_

Round 1
Reviewer 1 Report
I am quite sure that there are many studies about use of beneficial bacteria to control gray mould disease caused by Botrytis cinerea. However, I am not quite sure that the tested bacteria have plant growth-promoting activity. I suggest the authors re-define the topic.
Author Response
I am quite sure that there are many studies about use of beneficial bacteria to control gray mould disease caused by Botrytis cinerea. However, I am not quite sure that the tested bacteria have plant growth-promoting activity. I suggest the authors re-define the topic.
R: Dear reviewer, Indeed, there are several studies on microorganisms as biocontrol agents for the cause of the gray mold disease, Botrytis cinerea. Some of them are cited and reviewed in the work, highlighting their proposals. We even include other references suggested by the Editors. Hopefully, these changes are enough for you.
Reviewer 2 Report
this reviw is important for biological control of grey moluld fungus, in general, this text is well written, however, some issues should be addressed before it can be considered in this Journal, for example,
1 more detailed information such as mechanism should be included in Abstract section, the Abstract in present version is too general;
2 the authors summary some PGPB mechanisms but it seems these mechanisms are common for all antagonistic bacteria, the authors should give some new mechanisms such as molecular mechanisms or some specific mechamisms for grey mould fungus;
3 the authors descibed the disease cycle of grey mould, but we did not found the relationship between the disease cycle and biocontrol, then why introduce this disease cycle? if introduce, it should be helpful for the biological control of PGPB, or help us understant the molecular mechanisms of PGPB.
Author Response
this reviw is important for biological control of grey moluld fungus, in general, this text is well written, however, some issues should be addressed before it can be considered in this Journal, for example,
R: Thank you very much for your comments.
We are answering as follows:
1 more detailed information such as mechanism should be included in Abstract section, the Abstract in present version is too general;
R: You are right, thank you! The mechanisms are here in lines 23-25: ¨Some of PGPB mechanisms to control grey mould disease, include antibiosis, space occupation, nutrient uptake, ethylene modulation, and induction of plant defence mechanisms. In addition, recent studies on the action…¨
2 the authors summary some PGPB mechanisms but it seems these mechanisms are common for all antagonistic bacteria, the authors should give some new mechanisms such as molecular mechanisms or some specific mechamisms for grey mould fungus;
R: Thank you again for your suggestion; however, the biocontrol or antifungal mechanisms exerted by PGPB have been widely reviewed in other works, including molecular aspects. So, we did not want to repeat the same information. For example:
Blake, C., Christensen, M. N., & Kovács, Á. T. (2021). Molecular aspects of plant growth promotion and protection by Bacillus subtilis. Molecular Plant-Microbe Interactions, 34(1), 15-25.
Olanrewaju, O. S., Glick, B. R., & Babalola, O. O. (2017). Mechanisms of action of plant growth promoting bacteria. World Journal of Microbiology and Biotechnology, 33(11), 1-16.
3 the authors descibed the disease cycle of grey mould, but we did not found the relationship between the disease cycle and biocontrol, then why introduce this disease cycle? if introduce, it should be helpful for the biological control of PGPB, or help us understant the molecular mechanisms of PGPB.
R: The disease cycle produced by Botrytis cinerea helps the reader to understand some of the mechanisms carried out by PGPR to attack different growth stages, for example, the conidial and mycelial structures of the pathogen can be inhibited by PGPB compounds. This is mentioned in the abstract and in t text.
Reviewer 3 Report
The manuscript reviewed the stages of B. cinerea infection of host plants and the biocontrol mechanisms by PGPB. Oveerall, this paper povides important informations for biocontrolling the grey mould twith environmental friendly agricultural practices. However, the English writing should be improved before acceptable for publication. Please also see attachement for my comments.

Author Response
The manuscript reviewed the stages of B. cinerea infection of host plants and the biocontrol mechanisms by PGPB. Oveerall, this paper povides important informations for biocontrolling the grey mould twith environmental friendly agricultural practices. However, the English writing should be improved before acceptable for publication. Please also see attachement for my comments.
R: Thank you for your corrections. Some of them were also suggested by the Editors and modified accordingly. The work was also edited by a Native English speaker from Editage.com. In case you request a certificate we can prove it as evidence.
RESPONSE 2: Attached we provide the certificate of editing by Editage.com. The title then was different. Only that.

Round 2
Reviewer 1 Report
no comments
Author Response
no comments
Response: Thank you.
Reviewer 2 Report
this manuscript has been partially improved, but it need further improve due to short of novelty. Indeed, these antagonistic bacteria and antagonistic mechanism seems to be suitable for all plant diseases.
Author Response
this manuscript has been partially improved, but it need further improve due to short of novelty. Indeed, these antagonistic bacteria and antagonistic mechanism seems to be suitable for all plant diseases.
Response: We agree with you that PGPB is not specific or particularly antagonistic to Botrytis cinerea (nowhere we do not say the opposite). This has been widely reviewed and demonstrated by a bunch of works. What we try to resume in this work is the role of PGPB in the biocontrol of the grey mould fungus B. cinerea.
So, we would greatly appreciate it if you could be more specific with your suggestion to improve our work. What part of the review should be improved? In what way? What lines?
Reviewer 3 Report
The revised version is more informative with improved writing, and could be accepted for publication. However, minor modification still needed such as in line 667, the “speciessuch” should be “species such”. Please check through manuscript.
Author Response
The revised version is more informative with improved writing, and could be accepted for publication. However, minor modification still needed such as in line 667, the “speciessuch” should be “species such”. Please check through manuscript.
REPONSE: Modified as suggested, see new Line 236. All manuscript was double-checked, including references. All changes are highlighted. Thank you.
Round 3
Reviewer 2 Report
i could not understand why the authors emphasize this PGPR but indeed the manuscipt not give information about plant growth promotion, also not give a enough reason to justify for this PGPR,
also PGPR may control disease by modulation of soil microbial communicity
Author Response
i could not understand why the authors emphasize this PGPR but indeed the manuscipt not give information about plant growth promotion, also not give a enough reason to justify for this PGPR,
RESPONSE: Thank you for your comment. Changes are highlighted in the manuscript. We wrote the following paragraph to further clarify this issue. L84-90.
¨Some of the mechanisms of bacterial plant growth stimulation include providing plants with nutrients that are difficult to obtain directly from soil (or the atmosphere) such as nitrogen fixation, sequestering iron, or zinc and phosphate solubilization. The PGPB can also modulate the plant hormone levels, including indole-3-acetic acid, cytokinins, gibberellins or ethylene. The plant hormone ethylene has a wide range of biological activities and is synthesized as a response to various stresses [29,30]. ¨
We also included two recent references:
Khatoon, Z.; Huang, S.; Rafique, M.; Fakhar, A.; Kamran, M.A.; Santoyo, G. Unlocking the potential of plant growth-promoting rhizobacteria on soil health and the sustainability of agricultural systems. J. Environ. Manage. 2020, 273, 111118, doi:10.1016/j.jenvman.2020.111118.
Fadiji, A.E.; Santoyo, G.; Yadav, A.N.; Babalola, O.O. Efforts towards overcoming drought stress in crops: Revisiting the mechanisms employed by plant growth-promoting bacteria. Front. Microbiol. 2022, 13, 1–18, doi:10.3389/fmicb.2022.962427.
also PGPR may control disease by modulation of soil microbial communicity
RESPONSE: This an interesting topic, thank you. We inclueded teh following sentence and a new reference L509-511: ¨An interesting topic that would be good to delve into is the ability of PGPB to help plants recruit other beneficial microbiota, including those with the ability to restrict the growth of plant pathogens or pests [142]. ¨
Santoyo, G. (2022). How plants recruit their microbiome? New insights into beneficial interactions. Journal of advanced research. https://doi.org/10.1016/j.jare.2021.11.020